# The Interrelation between Interleukin-2 and Schizophrenia

**DOI:** 10.3390/brainsci12091154

**Published:** 2022-08-30

**Authors:** Yu Huang, Xin Zhang, Na Zhou

**Affiliations:** 1School of Pharmacy, Macau University of Science and Technology, Macau 999078, China; 2State Key Laboratory of Quality Research in Chinese Medicine, Macau 999078, China

**Keywords:** schizophrenia, interleukin-2, immune response, neuroinflammation biomarker

## Abstract

Interleukin-2 (IL-2) is a growth factor that regulates T-cell autocrine secretion and has long been considered to be closely related to immune response. With the advance in neuroinflammation theory and immunology research on schizophrenia, it is interesting and meaningful to discuss the possible role of IL-2 in schizophrenia. Here, we reviewed a series of studies published from the 1990s and found that IL-2 was closely associated with schizophrenia. For example, IL-2 is responsible for mediating toxic reactions, which are the causes of schizophrenia symptoms in patients, and such symptoms resolve after discontinuation of the drug. In addition, we focused on the changes of IL-2 in the onset, progression and treatment of schizophrenia and the possible mechanisms by which IL-2 affects schizophrenia. Our review suggests that IL-2 is associated with schizophrenia and plays a role in its pathogenesis, and progression IL-2 and sIL-2R could serve as potential biomarkers of schizophrenia.

## 1. Introduction

Schizophrenia is a severe chronic psychiatric disorder, with a lifetime prevalence of about 1% [1]. Clinically, the symptom of schizophrenia can be grouped into positive symptoms, (such as hallucinations and delusions), negative symptoms (such as social disorder and emotional passivation), mood symptoms (such as depressed mood), psychomotor symptoms, and cognitive impairment [2,3,4]. Most patients with schizophrenia suffer from the disease for life, with a low recovery rate. Based a meta-analysis, only 13.5% of patients with schizophrenia and related psychoses met recovery standard [5]. On the one hand, schizophrenia reduces the quality of life and causes a huge burden on patients and their family. On the other hand, schizophrenia reduces life expectancy. People with schizophrenia have an average life expectancy reduction of 15 years, and their suicide rate is between 5% and 10% [6]. The etiology of schizophrenia has remained unclear to date, and there is no appropriate animal model for preclinical study of the symptoms of schizophrenia. At present, dopamine hypothesis [7] and neuroinflammation theory [8] are widely recognized. The dopamine hypothesis is the earliest proposed pathomechanism of schizophrenia, which considers alterations of dopamine neurotransmission in the mesolimbic system that is responsible for positive symptoms and the mesocortical pathway, causing negative symptoms [9]. Meanwhile, neuroinflammation theory denotes that inflammation and infection of the central nervous system (CNS) can affect the onset of schizophrenia. For example, patients with schizophrenia and COVID-19 have a higher mortality rate than people with schizophrenia alone [10], which is possibly due to the inflammatory cytokine storm. In recent years, with the in-depth study of neuroinflammation theory of schizophrenia, the importance of inflammatory factors in schizophrenia has become increasingly prominent.

Interleukin-2 (IL-2) is a growth factor regulating the autocrine of T-cells (regulatory T, Treg) that plays a role in the immune response [11]. A large number of studies have shown that IL-2 levels are different in patients with schizophrenia when compared with the healthy control group [12,13,14]. This difference is thought to be related to inflammation. Lidia Yshii and others have found that IL-2 level is a limiting factor for Tregs in the human brain, and brain-specific IL-2 delivery is an effective strategy to suppress neuroinflammation [15]. The objectives of this review are to explore the possible impact of IL-2 on the prevalence and progression of schizophrenia, as well as the potential of IL-2 to be used as a marker of schizophrenia in order to provide novel insights for the treatment of schizophrenia.

## 2. IL-2, a T-Cells Regulatory Factor

IL-2 is a four helix cytokine that binds to IL-2 receptor (IL-2R) [16]. The IL-2R consists of three subunits:IL-2R α (CD25), IL-2R β (CD122) and γ c (CD132). The interaction of these subunits leads to the binding of IL-2 to cells. IL-2 is a comprehensive regulator in adaptive immune response that promotes lymphocyte proliferation and survival. In the immune system, IL-2 enhances the killing ability of natural killer cells, activates B cells to produce more immunoglobulins, maintains the stability of Treg cells and regulates the differentiation of effector T-cells. IL-2 is also involved in activating CD4+ and CD8+ T cells, inducing immune response [17]. Activated CD4+ T cells secrete an increased amount of IL-2 to enhance the expression of IL-2Rα, improve the affinity of IL-2, and promote the proliferation and differentiation of T cells after antigen stimulation. IL-2 also plays a key role in CD8+ T cells. After acute infection, naïve antigen-specific CD8+ T cells proliferate and differentiate into cytotoxic T lymphocytes (CTLs). These cells can produce cytotoxicity to the infected target cells and release cytokines, such as IFN- γ [18] and TNF- α [19], to eliminate pathogens. After TCR stimulation, IL-2 can drive the expansion of CD8+ T-cell population and enhance the memory bank of CD8+ T cells. High levels of IL-2 signaling drives T-cell differentiation and promotes the expression of key cytolytic effector molecules and cytokines through immunoactivated CD8+ T cells [20]. Low IL-2 treatment can stimulate the maturation of CD8+ T memory cells, and the upregulation of IL-2R α level in the early stage after TCR stimulation, which is critical for the formation of memory cells [21,22,23].

Several studies have shown that IL-2 has an important effect on Treg cells. Liu et al. observed that the aggregation of active effector T cells could produce IL-2 around pSTAT5 + Treg cells [24]. This finding supported the role of IL-2 secreted by T cells in maintaining the stability of Treg cells in lymphoid organs and tissues. An in vivo study using mouse models demonstrated that there was a significant defect in the development of Treg cells in thymus and a significant decrease in Treg cells in peripheral lymphoid tissue without T-cell-derived IL-2 [25]. Evidence indicated that Treg cells have higher and more sustained IL-2 signaling than that of other T-cell types [26]. Furthermore, studies have shown that IL-2 is necessary for the survival, proliferation, functional capacity and activation of Tregs expressing CD4+/CD25+/Foxp3+ and is likely to be involved in the generation in the thymus. However, the imbalanced distribution of IL-2 may cause Tregs instability, Foxp3 reduction and cytokines’ release that exacerbate autoimmunity [27]. In humans, Treg cells have special transcriptional amplification mechanism. For example, low-dose IL-2 selectively activates Treg cells by activating IL-2/IL-2R signaling in type 1 diabetes [26]. In addition, other aspects of the role of IL-2 need to be considered in future studies, including the dual role of IL-2 in Treg cell activation, the balance between Teff and Treg cells, and therapeutic strategies such as IL-2 monoclonal antibodies. These may enhance our understanding of the role of IL-2 in immunity.

## 3. Schizophrenia and Neuroinflammation

The immune and nervous systems are two important regulatory systems to maintain the internal environment of the body. The nervous system controls the function of the immune system, and the immune system can regulate the nervous system through cytokines. Recent studies have shown that the immune inflammatory response of the brain plays a key role in the pathogenesis of several central nervous system diseases, including schizophrenia. Although the pathogenesis of schizophrenia is not well established, dopamine has been considered to be related to schizophrenia for a long time, with ample preclinical evidence supporting this hypothesis [28,29,30]. However, the mechanism of how dopamine causes schizophrenia remains unclear, and the therapeutic effect of antipsychotic and anti-dopamine drugs has not been satisfactory in clinical treatment. With the advance in research on schizophrenia, the theory of neuroinflammation has become increasingly recognized. The vulnerability-stress model, first proposed in the last century, emphasizes the role of stress in the onset of schizophrenia [31]. In addition, the vulnerability-stress-inflammation model proposed by Müller indicates that inflammation plays an important role in the pathogenesis of schizophrenia [8]. The mechanism of neuroinflammation involved in the pathophysiological process of schizophrenia is complex, and it mainly affects glutamate metabolism, inducing oxidative stress, neuronal apoptosis, etc. Inflammation of the CNS is usually mediated by microglia, cytokines, astrocytes and immune cells (such as T cells). Microglia cells are the immune cell in the CNS, accounting for about 15% of the total CNS cells [32]. Microglia cells are sensors of cytokines, which play an important regulatory role in the CNS and are also the first line of defense against CNS infection [33]. At the end of the 20th century, Bayer et al. found that activated microglia in the frontal cortex and hippocampus increased significantly in 14 patients with schizophrenia when compared to 13 normal subjects [34]. Similar studies reported that microglia in dorsolateral prefrontal cortex, upper-temporal-lobe gray matter and anterior cingulate gyrus gray matter were significantly activated in patients with schizophrenia [35]. Uranova et al. observed the increase of activated microglia in the brain of patients with schizophrenia by electron microscope [36]. In addition to microglia, cytokines also have a great impact on schizophrenia. There have been many studies on the changes of cytokine levels in patients with schizophrenia. No marker for the diagnosis, treatment or disease progression has been established for schizophrenia so far. Cytokines are some of the key candidates in this research field, and Patel et al. have published a comprehensive review on this topic [37]. 

Infection and immunity have long been the focus of research on the etiology of schizophrenia. Intrauterine infection can increase the risk of mental illness and is often associated with early events in brain development [38]. A number of studies have found that there is an increased incidence of schizophrenia in people who were prenatally exposed to viral and bacterial infections and Toxoplasma gondii [39,40,41]. Most previous studies have been based on epidemiological surveys to find correlation between prenatal infection and neurodevelopmental outcomes in offspring [42]. However, with further research, the underlying mechanism of such correlation is becoming clearer. Exogenous infections can have direct and indirect effects on the fetal brain, resulting in varying degrees of neurological damage. The classic TORCH is an abbreviation for a type of infection that can cause severe brain malformation in fetuses and increase their risk of developing mental disorders [43]. In addition, maternal immune activation (MIA) may affect fetal brain development, thereby increasing their risk of developing schizophrenia. Studies suggest that maternal immunity will release inflammatory mediators after activation through maternal blood or amniotic fluid and affect fetal CNS development [44]. MIA can induce a variety of cytokine changes in the fetal brain, but the effect of these changes on the fetal brain is still poorly understood [45,46]. Interestingly, in animal studies, the offspring of mothers injected with IL-2 developed schizophrenia-like symptoms [47].

## 4. H-Treg Hypothesis in Schizophrenia

The Treg cell is the main cell type in maintaining immune homeostasis, including the innate and adaptive immune balance. The important role of IL-2 in the production of Treg cells has previously been described. In CNS, Tregs mainly regulate the neuroimmune interaction between astrocytes and microglia [48]. Impaired Treg cells induce astroglial overactivation and microglial pruning in schizophrenia [49]. Foxp3 is the main regulator of Treg cells and is very important for their development and inhibition [50]. Reduced CD4+/CD25+/Foxp3+-expressing Tregs with increased proinflammatory response in patients with schizophrenia were reported by Corsi-Zuelli last year [51].

Furthermore, Corsi-Zuelli et al. proposed the H-Treg hypothesis, which states that decreased Treg function could provide a missing link between low-level systemic inflammation and abnormal central glial regulation in psychosis [49]. The reason for supporting this hypothesis is that studies found that low-dose methotrexate improved the positive symptom scores in patients with schizophrenia when compared to a placebo group [52]; this result may be due to reset systemic regulatory T-cell control of immune signaling. These results suggest that restoring Tregs’ function might be a novel therapeutic target in schizophrenia.

## 5. Possible Effects of IL-2 in Schizophrenia

The effect of IL-2 has been widely studied in different aspects of schizophrenia. As early as the last century, studies had suggested a correlation between IL-2 and schizophrenia. IL-2 can affect many pathological processes which are related to the pathogenesis of schizophrenia, such as the Glu neurotransmitter system. Studies have shown that IL-2 can induce paroxysmal firing in the hippocampal structure of rats and affect the synthesis and release of Glu and GABA in hippocampal neurons. In addition, in vitro studies have shown that IL-2 can cause an increase in Glu immune response neurons and a decrease GABA in immune-response neurons [53]. IL-2 exerts its biological effect by binding the IL-2 receptor on the cell membrane. It acts not only in the immune response of the body, but also in some physiological or pathological processes of the CNS as a neural mediator [54]. Denicoff et al. performed systematic treatment of IL-2 activated killer cells in 44 patients with metastatic cancer [55]. The results showed that, among the 44 patients, 15 had serious behavioral changes, and half of the patients had serious cognitive changes. IL-2 had a dose–response relationship with neuro-behavioral changes. In an study, at a low dose, 4 of 19 patients showed schizophrenic symptoms, while at a high dose, 10 of 24 patients showed symptoms, and the symptoms of all patients disappeared after stopping treatment [56]. The mechanism for this effect has not been explored, but it might be due to IL-2-mediated cytotoxicity. Furthermore, Schwarz et al. analyzed the relationship between IL-2 and susceptibility genes for schizophrenia and found that there was a significant association between the IL-2–330TT genotype and schizophrenia in 230 patients with schizophrenia (χ^2^ = 7.418, df = 2 and *p* = 0.024) [57]. According to previous in vitro studies, IL-2–330TT genotype polymorphism was shown to be related to IL-2 level [58]. Another study explored the relationship between IL-2RG gene and schizophrenia. Overall, mRNA expression was detected in the peripheral blood of 66 patients with schizophrenia and 99 healthy subjects. The results showed that, compared with the control group, the expression of IL-2RG mRNA in patients’ peripheral blood was upregulated (patients vs. controls, median [interquartile range]: 2.080 [3.428–1.046] vs. 0.324 [0.856–0.000], *p* < 0.0001), suggesting that the altered immune response in schizophrenia might be related to the overexpression of IL-2RG [59]. Taken together, these studies suggest a potential genetic link between IL-2 and schizophrenia.

Extensive studies have been carried out on the change of the IL-2 level in patients with schizophrenia. However, the results of these studies vary largely, with some studies showing an increase in the level of IL-2 in in patients with schizophrenia [12,13,14], while some found a decrease [60,61] or no change [62]. There seems to be no generally accepted explanation for this inconsistency, and many possible reasons could contribute to it that may be related to race, disease time, disease process, disease severity and the selection of control. Among these reasons is the fact that the disease process affects the stability of IL-2 in patients with schizophrenia. It is generally believed that IL-2 changes significantly in patients with acute attack, while it tends to stabilize in patients in a stable stage. In addition, the choice of experimental methods is also highly relevant. Errors or undetected states could occur due to the low level of IL-2 in human body and the high detection limit of general detection methods. Despite inconsistent results on IL-2, changes in IL-2R and soluble interleukin-2 receptor (sIL-2R) levels in patients with schizophrenia are also noteworthy. The level of sIL-2R in patients with schizophrenia is consistently higher than that in healthy controls across different studies; therefore, it could potentially serve as a diagnostic marker of schizophrenia [63,64,65,66].

IL-2 seems to be particularly associated with negative and cognitive symptoms of schizophrenia. Asevedo compared possible correlations between peripheral IL-2 level and symptoms and cognitive function in patients with schizophrenia [60]. When comparing 29 outpatients with chronic medication who received schizophrenia treatment with 26 healthy controls, IL-2 level was found to have negative correlation with the degree of negative symptoms and language disorder. A real-world study showed that the IL-2 level is a key factor in predicting negative symptoms and cognitive impairment in outpatients with schizophrenia [67]. So far, few biomarkers have been reported in relation to schizophrenia symptoms, and no accurate conclusions have been formed. However, IL-2 seems to be a symptom-related markers, which may provide new auxiliary evidence for predicting the severity of clinical symptoms.

The treatment of schizophrenia remains a great challenge. The mainstream antipsychotics drugs available are not effective in regard to all symptoms of schizophrenia. Basically, all antipsychotic drugs on the market are based on the dopamine D2 receptor. The first- and second-generation antipsychotics are dopamine D2 receptor antagonists, while the third-generation drugs are partial agonists or partial ligands of this receptor, and others are antagonists of the similar D2 receptor (D3D4D5) [68].The representative drug of the first generation of antipsychotics is chlorpromazine, which has been marketed since 1953 and was the first antipsychotic drug to work mainly by blocking D2 receptors in the brain. However, its lack of selectivity led to a wide range of off-target side effects. The most common and serious side effects of chlorpromazine include extrapyramidal effects, dyskinesia, dystonia, involuntary movement and so on. The representative drug of the second-generation antipsychotics is clozapine. Compared with the first-generation drugs, the second-generation antipsychotics have a stronger ability to block serotonin 5-HT2A receptors than dopamine D2 receptors; therefore, they have a stronger effect on negative symptoms. The second-generation antipsychotics also have a weaker antagonistic effect on D2 receptors and therefore have a lower extrapyramidal effect. Aripiprazole represents the third and latest generation of antipsychotics. The key difference between aripiprazole and the previous two generations is that it is not a D2 receptor antagonist, but a partial agonist. It can regulate the level of dopamine in the body and keep it in a relatively stable state. Therefore, it is also called “dopamine stabilizer” [69]. Giridharan found that clozapine could affect the level of IL-2 and exhibit an important inhibitory effect on inflammation due to its ability to inhibit the expression of NLRP3 inflammatory body and to play an anti-inflammatory role by affecting the clozapine precautions poly (I: C) pathway [70]. An in vivo study in adolescent mice showed that IL-2 injection led to a significant change in stereotypical behavior [71]. A potential explanation of this effect might be that IL-2 affects the pruning process, in which dopaminergic synapses and receptors are eliminated. This also suggests that there may be a relationship between IL-2 and dopamine, which affected the therapeutic effects of antipsychotics. In addition to animal experiments, some human experiments have proved that IL-2 affects the therapeutic effect of schizophrenia or that the IL-2 level changes after antipsychotics treatment. A meta-analysis has shown that IL-2 levels decreased in patients with schizophrenia who were treated with antipsychotics [72]. Yuan et al. found that the IL-2 level in patients with refractory schizophrenia after clozapine treatment was positively correlated with the dose of clozapine [73]. Interestingly, such a correlation seems to exist in females only.

## 6. Conclusions

So far, the etiology of schizophrenia remains unclear, with the dopamine hypothesis being the most recognized theory [74]. However, the dopamine hypothesis cannot explain all symptoms, and neuroinflammation is increasingly recognized as a supplementary cause. At present, there is still a lack of effective markers for diagnosis and treatment process of schizophrenia. Although several studies have attempted to establish various cytokines as markers for schizophrenia (such as protein-C Reactive and BDNF), there were always deficiencies when applying these markers in the clinical practice [75,76]. As a T-cell regulatory factor, IL-2 plays a key role in inflammation and immunity. The role of IL-2 in schizophrenia was discussed in this review. IL-2 level is correlated with the pathogenesis and treatment response of schizophrenia, with ample evidence from both vitro and in vivo studies. However, further prospective validation is needed to define the molecular mechanisms through which IL-2 influences schizophrenia initiation and development. There is no antipsychotic drug developed with IL-2 as target, and this could also become a research direction in the future. Moreover, the levels of IL-2 in patients with schizophrenia often differ from healthy controls. IL-2 level is closely related to negative symptoms and cognitive deficit and is also varied after treatment. There are higher levels of sIL-2R untreated schizophrenia patients without therapy. Taken together, our findings suggest that IL-2 or sIL-2R may be potential biomarkers for schizophrenia. Further research is needed to validate this hypothesis.

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
