# Peer review of "The Interrelation between Interleukin-2 and Schizophrenia"

_brainsci, 2022, doi:10.3390/brainsci12091154_

Round 1

Reviewer 1 Report

The manuscript reviews the literature looking at the relationships between IL2 and schizophrenia. The aim is interesting. However, I have some comments for the authors that need to be addressed:

- line 19: schizophrenia is a psychiatric disorder more than a neurological disease. I agree that there is several structural evidence of alteration if compared to the general population, but the sentence needs to be revised. 

- there are papers about the relationship between inflammatory and brain development in schizophrenia, as a possible mechanism for the disorder. Please include this aspect in your review because it is an important one, especially thinking at the high risk population (for example see Buka et al., 2001 https://doi.org/10.1006/brbi.2001.0644, or Wu et al., 2019 https://doi.org/10.1016/j.brainres.2019.146463). I understand that it is not a systematic review, but it is not possible to neglect an important part of the literature. 

- it is unclear to me if the authors have a hypothesis that could be drafted from their review. It could be interesting for future study or discussion in the field.

- have you evaluated the possibility of including a figure? it could be more helpful in the understand of your paper

- There are several typing errors in the manuscript. Please, revise them (lines 23, 27, ..., 63, ...). However, I think the manuscript might be also revised by a Native speaker. 

Reviewer 2 Report

This is an interesting paper about the association between interleukin-2 and schizophrenia. The authors conducted a review on the topic. The paper is well-written and interesting for the readers and the journal; however, several changes should be made before considering it for publication.

Abstract

1-The abstract should be structured. The authors conducted a narrative review. Which methods did they use? Can the authors report in the abstract which search terms did they use?

2-What are the main findings of the review? Are the authors comparing these findings with those found in other inflammatory markers?

Introduction

1- The authors are reporting that schizophrenia is a severe neurological disease. I prefer to name it as a mental disorder, not a neurological disease.

2-Clinical domains in schizophrenia are not only positive, negative and cognitive. What about the depressive and other domains? Can the authors add more new references?

3-At the end of the introduction section, the authors should expand the main aims of the study. A subsection called "aims" would be a good option.

Methods

1- There is no material and methods section. How did the authors conducted the review? Is it a narrative review? Did they use SARA guidelines?

Results

1- Results should not only be restricted to the Il-2. I recommend to support their findings with others from the inflammatory hypotheses.

Discussion

1- A summary section is not needed and should not be provided. The correct form is to perform a discussion section, with a subsequent conclusions section.

Round 2

Reviewer 1 Report

I think the authors have addressed all my concerns and it is suitable for publication